# Genomic GC bias correction improves species abundance estimation from metagenomic data

Laurenz Holcik [1], Arndt von Haeseler [1,2] & Florian G. Pflug [1,3] ✉

Metagenomic sequencing measures the species composition of microbial communities and has revealed the crucial role of microbiomes in the etiology of a range of diseases such as colorectal cancer. Quantitative comparisons of microbial communities are, however, affected by GC-content-dependent biases. Here, we present GuaCAMOLE, a computational method to detect and remove GC bias from metagenomic sequencing data. The algorithm relies on comparisons between individual species in a single sample to estimate the sequencing efficiency at levels of GC content, and outputs unbiased species abundances. GuaCAMOLE thus works regardless of the specific amount or direction of GC-bias present in the data and does not rely on calibration experiments or multiple samples. Applying our algorithm to 3435 gut microbiomes of colorectal cancer patients from 33 individual studies reveals that the type and severity of GC bias vary considerably between studies. In many studies, we observe a clear bias against GC-poor species in the abundances reported by existing methods. GuaCAMOLE successfully removes this bias and corrects the abundance of clinically relevant GC-poor species such as *F. nucleatum* (28% GC) by up to a factor of two. GuaCAMOLE thus contributes to a better quantitative understanding of microbial communities by improving the accuracy and comparability of species abundances across experimental setups.

Metagenomic sequencing has enabled the comprehensive and quantitative analysis of taxa abundances in a wide range of microbial communities[1]. It has uncovered the importance of microbiomes amongst others in health, disease, nutrition and ecology, and has revealed a complex interplay between microbial consortia and their hosts[2–4].

Metagenomic sequencing relies on comprehensive high-throughput sequencing of all DNA in a sample to quantify the abundance of all present taxa. To prepare a sample for sequencing, the DNA is extracted, purified, fragmented, amplified, and finally outfitted with sequencing adapters. Numerous protocols have been established for these library preparation steps, each differing in methods and materials used[5–7]. After sequencing, the reads are assigned to taxa, and relative read counts are used as proxies for the taxa's abundances[8,9]. This assumes that reasonably accurate genomes of all species present in the sample are available. Alternatively, unknown genomes can in principle be assembled from individual reads through a process called metagenomic assembly[10–13]. Metagenomic assembly, however, has a bias profile very different from that of read assignment, so we do not consider this case further here.

While metagenomic sequencing is in principle agnostic to the specific taxa in a sample, library preparation can introduce sequence-dependent biases.[14] In particular, the GC content (i.e., fraction of G and C bases in the sequence) has been shown to strongly affect sequencing efficiency[15]. Metagenomic sequencing is

[1]Center for Integrative Bioinformatics Vienna (CIBIV), University of Vienna, Vienna, Austria. [2]Ludwig Boltzmann Institute for Network Medicine, University of Vienna, Vienna, Austria. [3]Biological Complexity Unit, Okinawa Institute of Science and Technology, Onna, Okinawa, Japan. ✉e-mail: florian.pflug@oist.jp

particularly affected because the genomic GC content often differs significantly between species[5]. The magnitude and even direction of this bias, however, vary between different library preparation and sequencing protocols[16]. For example, a low GC content can either increase or decrease sequencing efficiency depending on the precise protocol used. As a result, computational correction for GC bias has been challenging[17].

The species on the extreme ends of the genomic GC content range are particularly prone to biases. Amongst these species, we find pathogenic taxa such as *F. nucleatum* (28% GC content, associated with colorectal cancer) and *M. pneumoniae* (25% GC content, associated with pneumonia)[18–20]. With common sequencing protocols, the abundance of these taxa will be underestimated[5,17], and this can affect even comparisons between samples analyzed using the same protocol[21].

Ideally, GC bias should therefore be removed on a per-sample level. Computational methods to remove GC bias have been developed for various sequencing-based methods, and have been shown to be crucial to avoid skewed results[15,22]. These methods, however, assume reads are aligned to a reference genome. For metagenomic samples possibly containing thousands of taxa, creating such an alignment is prohibitively computationally expensive. Instead, metagenomic reads are typically assigned to taxa using k-mer-based methods[8,23–25]. This makes existing methods inapplicable and requires a novel approach to GC bias correction.

We present the GuaCAMOLE (Guanosine Cytosine Aware Metagenomic Opulence Least Squares Estimation) algorithm for the efficient detection and removal of GC bias from metagenomic samples. GuaCAMOLE is an alignment-free algorithm and instead assigns reads to taxa using Kraken2[8]. The algorithm also does not require calibration data or any a priori assumptions about the quantitative relationship between GC content and bias (such as extremely GC-rich and GC-poor species being more prone to bias), and thus works equally well for all sequencing protocols.

Using both simulations and experimental data[16], we show that GuaCAMOLE uncovers protocol-specific GC bias and improves abundance estimates over existing methods. To show that GC bias correction can be relevant in a clinical setting, we apply GuaCAMOLE to a large number of metagenomic stool samples of colorectal cancer patients[26–28]. Here, we observe that the type and severity of GC bias varies strongly between studies, and that accounting for GC bias significantly increases the estimated abundances of clinically relevant taxa on both ends of the GC spectrum.

## Results

The GuaCAMOLE algorithm processes the raw sequencing reads of a metagenomic sample and outputs bias-corrected abundances for all detected taxa. GuaCAMOLE also infers and outputs GC-dependent sequencing efficiencies, which reflect the probability (relative to the maximum) that a DNA fragment with a certain GC content successfully undergoes all library preparation steps and sequencing. These GC-dependent sequencing efficiencies thus measure the extent of the GC bias present in the raw data. Briefly, GuaCAMOLE works as follows (Fig. 1, see Methods for details): Reads are first assigned to individual taxa using Kraken2[8], and within each taxon to discrete bins representing the read's GC content (these bins are subsequently referred to as taxon-GC bins). Reads which cannot be assigned to a specific taxon unambiguously by Kraken2 are redistributed probabilistically to the likeliest taxon using the Bracken algorithm[29]. Read counts in each taxon-GC-bin are then normalized based on expected read counts computed from the genome lengths and genomic GC content distributions of individual taxa. The resulting quotients only depend on the unknown abundances (one for each taxon) and unknown GC-dependent sequencing efficiency (one per GC-bin). From these quotients, we then compute bias-corrected abundance estimates and the GC-dependent sequencing efficiencies. GuaCAMOLE reports the estimated abundances either as *sequence abundances* proportional to the total amount of DNA present, or *taxonomic abundances* proportional to the number of genomes[30].

### GuaCAMOLE improves accuracy on simulated communities

We first demonstrate that GuaCAMOLE infers the correct abundances and GC-dependent sequencing efficiencies independent of the specific type of GC bias present. We ran GuaCAMOLE on data simulated using three different models of GC bias (see Methods for details): peak efficiency at 50% GC, efficiency increasing with GC content, and efficiency decreasing with GC content (Fig. 2A). For all three simulated datasets, GuaCAMOLE produced virtually unbiased estimates (mean relative error less than 1%) and correctly recovered the GC-dependent sequencing efficiencies used for the simulation. The Bracken estimates showed considerable GC bias in comparison (relative errors 10% to 30% depending on the bias model).

We next confirmed that GuaCAMOLE performs well for metagenomic communities with different complexities and species compositions (Fig. 2B). We simulated sequencing libraries representing communities comprising different numbers of taxa (5, 10, 50, 100, or 400 taxa) with log-normally distributed abundances. Taxa were

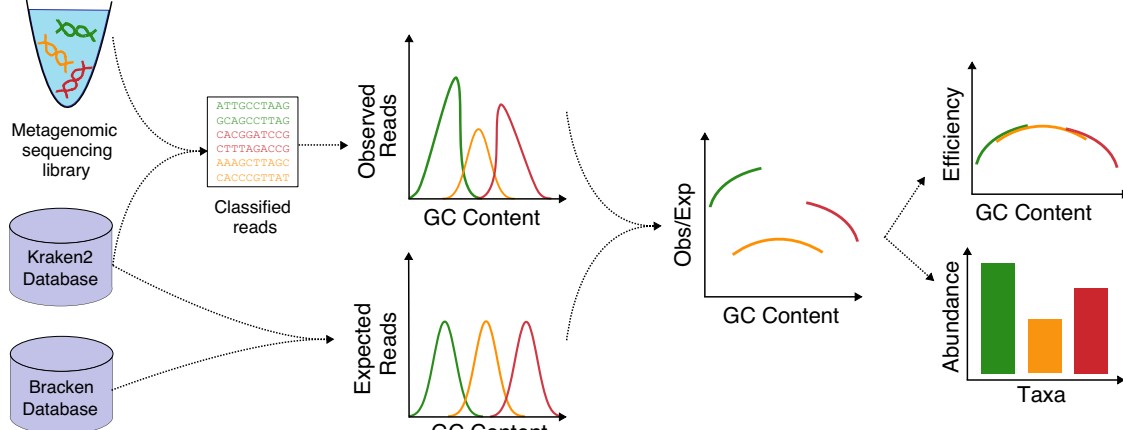

**Fig. 1 | The GuaCAMOLE algorithm.** Reads are assigned to taxa using Kraken2/Bracken[8,29] and binned into discrete GC bins per taxon. Corresponding expected read counts are then computed for each taxon and GC bin from the reference genomes. The observed/expected quotients reflect the GC-dependent sequencing efficiencies scaled by each taxon's abundance. Abundances are estimated by finding the scaling factors for which the quotients form a continuous curve.

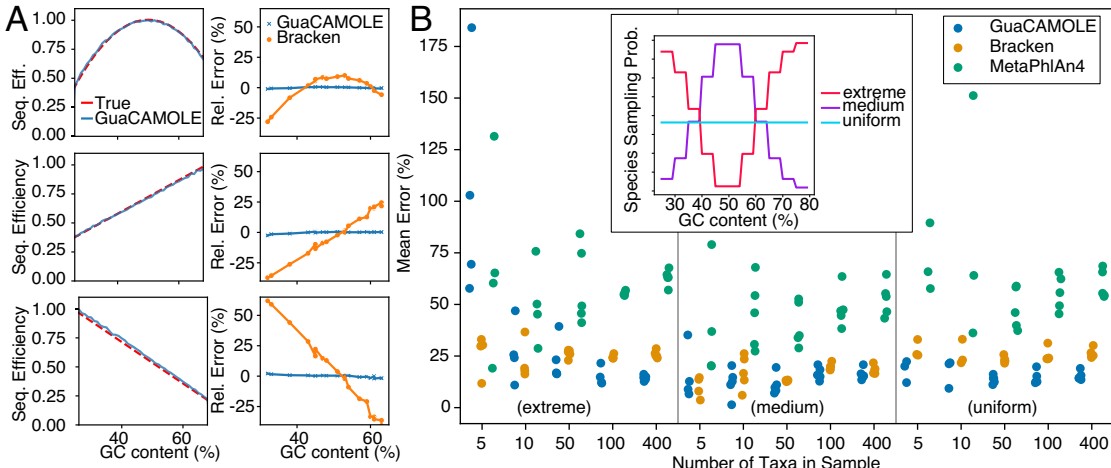

**Fig. 2 | Performance on simulated metagenomic data. A** Three qualitatively different types of GC-dependent sequencing bias (left column; true bias in red and GuaCAMOLE estimate in blue) and the corresponding abundance estimation errors (right column) of GuaCAMOLE (blue) and Bracken (orange). **B** Mean relative estimation error of GuaCAMOLE, Bracken, and MetaPhlAn4 for 5 simulated communities for each combination of size (5, 10, 50, 100, and 400 taxa) and genomic GC content distributions across taxa (extreme, medium or uniform; see inset). Replicates for which GuaCAMOLE failed to produce an estimate are omitted. To avoid overlaps, dots are horizontally shifted (GuaCAMOLE left, MetaPhlAn4 right) and jittered.

chosen from the RefSeq database, and selected to either have predominantly low or high genomic GC content (extreme), predominantly GC content around 50% (medium), or a GC content uniformly distributed over the whole range (uniform). For each combination of community complexity and distribution of GC content, we simulated 5 libraries with a GC-dependent sequencing efficiency of $1 - 10 \cdot (g - 0.5)^2$ so that the efficiency of $g = 30\%$ and $g = 70\%$ GC content was 40%. We then compared the relative estimation errors of GuaCAMOLE with those of the other RefSeq-based algorithms, Bracken and MetaPhlAn4. For libraries comprising 50 taxa or more, we find that GuaCAMOLE consistently shows the lowest mean estimation error (Fig. 2B). As expected, the advantage over other algorithms is the largest for communities that predominantly contain taxa with extreme GC content (extreme). When species mostly have a GC content around 50% (medium) Bracken and GuaCAMOLE perform similarly. For small communities comprising mostly taxa with extremely high or low GC content, the accuracy of GuaCAMOLE is severely reduced. For 8 out of 75 simulated communities (4 comprising 5 taxa, 3 comprising 10 taxa, one comprising 50 taxa), GuaCAMOLE was unable to produce reliable estimates and exited with a warning. The likely cause is that for these communities, the GC content distributions of individual taxa did not overlap sufficiently. Since this occurs mainly when all taxa have extreme GC content, we expect this case to be rare in practice.

**Improved accuracy across a range of experimental protocols**

Having tested GuaCAMOLE on simulated data, we went on to show that it improves abundance estimates for experimental data produced using different protocols, and that GuaCAMOLE can uncover the GC-dependent sequencing efficiencies of these protocols (Fig. 3). We re-analyzed data published by Tourlousse et al.[16] of a mock community sequenced using 28 different protocols (Table 1) with GuaCAMOLE, Bracken[29], MetaPhlAn4[9], SingleM[31], Sylph[32] and mOTUS[33]. The mock community comprises 19 bacterial species representative of human-associated microbiota and was sequenced using 11 different commercially available library preparation kits (labeled A-K below, see Table 1). For each kit, Tourlousse et al. tested up to three PCR amplification regimes: 500 ng input DNA with no PCR amplification (suffix 0), 50 ng input DNA with 4-8 PCR cycles (suffix L), and 1 ng input DNA with 8-15 PCR cycles (suffix H).

We find that the GC-dependent sequencing efficiencies estimated by GuaCAMOLE differ strongly between different protocols (Fig. 3A).

Some protocols show uniform efficiencies, while others show a strong dependence on the GC content. In accordance with the results of Tourlousse et al.[16] we see that the protocols DH, FH, GH, IL, and IH (see Table 1) show the strongest dependency on GC content (Fig. 3A, colored lines). The nature of this dependence differs qualitatively between protocols. While protocols IL and IH show decreasing sequencing efficiency with increasing GC content, DH, FH, and GH show an increase in efficiency for higher GC content.

For the protocols most strongly affected by GC content (DH, FH, GH, IH, IL) GuaCAMOLE reduces the mean relative abundance error drastically compared to the other algorithms (Fig. 3B, colored dots). For other protocols GuaCAMOLE and Sylph overall show the smallest error, with a considerably larger variation of errors for Sylph than for GuaCAMOLE (Fig. 3B). Looking at individual protocols, GuaCAMOLE and Sylph show the smallest estimation errors for all protocols except protocol JH (Fig. 3C). A common source of GC bias is PCR amplification[34], and accordingly the advantage of GuaCAMOLE over the other algorithms increases with the number of PCR cycles (Fig. 3D). However, GuaCAMOLE also offers a clear advantage over the other algorithms for PCR-free protocols A0 and C0, (Fig. 3C).

The quantification error per bacterial species shows for GuaCAMOLE only a weak residual dependence on genomic GC content (Fig. 3E). In comparison, the quantification error of the other tested algorithms increases significantly for taxa on the extremal ends of the GC content range. The runtime of GuaCAMOLE (including the runtime of Kraken2 itself) is slightly longer than most other algorithms (Table S1), but the difference of up to 2x makes GuaCAMOLE still a practical choice.

The choice of protocol strongly affects GC-dependent sequencing efficiencies also in other datasets. For a human gut mock community[35] comprising 18 taxa sequenced on two different sequencing platforms, GuaCAMOLE reveals that the sequencing platform (Illumina HiSeq 2500 rapid and NovaSeq 6000 SP) has a significant influence (Supplementary Fig. S1). Single-species libraries containing only *Fusobacterium sp. C1*[17] yields similar results (Supplementary Fig. S2).

**GC-dependent sequencing efficiency differs widely between studies**

To test how much GC-dependent sequencing efficiencies affect real-world studies, we ran GuaCAMOLE on 3435 samples from 33 studies of human gut microbiomes of healthy patients and patients with

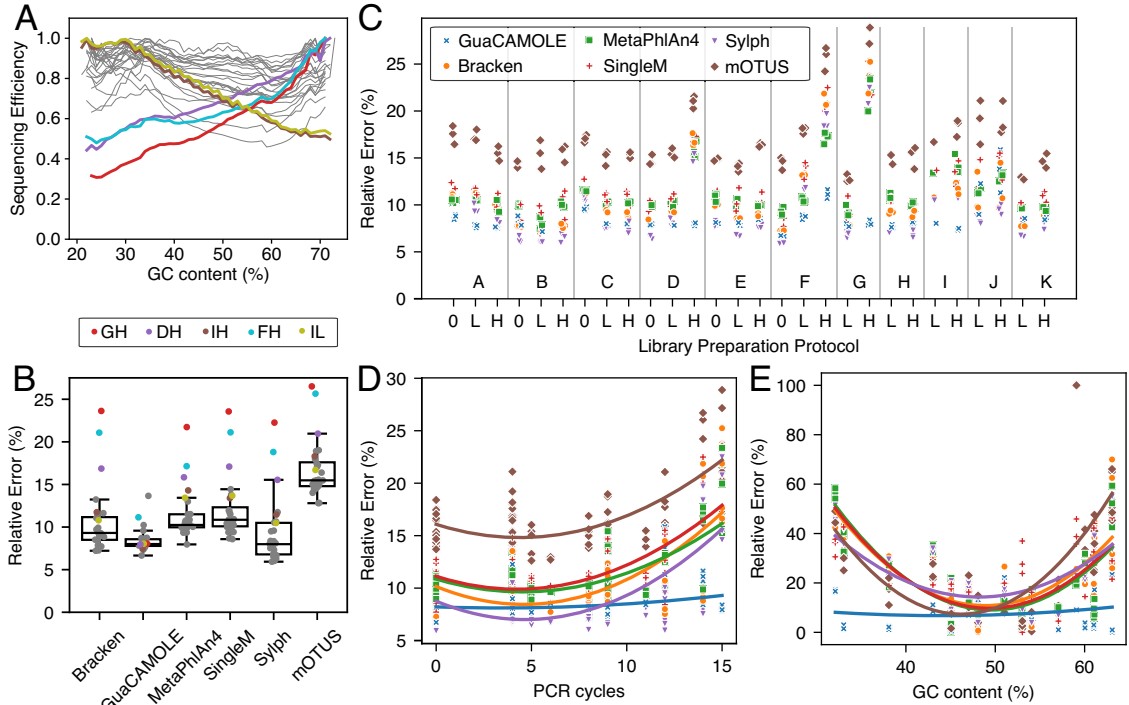

**Fig. 3 | Performance of GuaCAMOLE for experimental mock community data.**
The mock community of Tourlousse et al.[16] contains 19 species and was sequenced in triplicate using 28 different library preparation protocols (Table 1). We re-analyzed the reads with GuaCAMOLE, Bracken, MetaPhlAn4, SingleM, Sylph and MOTUS. GuaCAMOLE was set to report taxonomic abundances, Bracken results were manually adjusted for genome length. Relative estimation errors are $\frac{1}{19}\sum_j |a_j - A_j|/A_j$ where $A_j$ is the true and $a_j$ the estimated abundance of species $j = 1, ..., 19$. **A** Estimated GC-dependent sequencing efficiencies of the 28 protocols by GuaCAMOLE. Highlighted protocols GH, DH, IH, FH, IL were found by Tourlousse et al. to exhibit the strongest dependency of efficiency on GC content. **B** Relative estimation errors per protocol, averaged over all replicates. Protocols GH, DH, IH, FH, IL are highlighted, see (A). Each boxplot represents 19 datapoints (one per protocol), and shows the median (center line), 25% and 75% quantiles (hinges) and the furthest point less than 1.5 IQRs (inter-quartile ranges) from the nearest hinge. **C** Relative estimation errors for the three replicates of each protocol. **(D)** Relative estimation error vs. number of PCR cycles used in each protocol. Lines show quadratic best fit, colors indicate the algorithm as in (**C**). **E** Relative abundance estimation error of each taxon averaged across protocols vs. genomic GC content. Lines show quadratic best fit, colors indicate the algorithm as in (**C**).

**Table 1 | DNA Library Preparation Kits tested by Tourlousse et al.[16]**

| kit name | fragmentation method | PCR cycles (O/L/H) | abbr. |
|---|---|---|---|
| Accel NGS 2S Plus DNA Library Kit | physical | 0/4/9 | A O/L/H |
| QIAseq FX DNA Library Kit | enzymatic | 0/8/12 | B O/L/H |
| TruSeq (Nano) DNA (PCR-Free) Library Kit | physical | 0/8/8 | C O/L/H |
| KAPA HTP Library Preparation Kit | physical | 0/5/15 | D O/L/H |
| KAPA HyperPrep (PCR-free) Kit | physical | 0/4/14 | E O/L/H |
| KAPA HyperPrep + KAPA Frag | enzymatic | 0/4/14 | F O/L/H |
| KAPA HTP + KAPA Frag | enzymatic | −/5/15 | G L/H |
| NEBNext Ultra II DNA Library Prep Kit | physical | −/4/9 | H L/H |
| NEBNext Ultra II FS DNA Library Prep Kit | enzymatic | −/4/9 | I L/H |
| Nextera DNA Flex Library Prep Kit | enzymatic | −/4/12 | J L/H |
| SMARTer ThruPLEX DNA-Seq Kit | physical | −/6/11 | K L/H |

Suffices O/L/H represent no PCR amplification (0), low number of PCR cycles (L; 50 ng input DNA), and high number of PCR cycles (H; 1 ng input DNA).

colorectal cancer (CRC), selected according to sample quality[27,28]. 3031 of those samples were usable and reported reliable sequencing efficiencies (less than 50% reads assigned to taxa classified as false positives). By clustering these 33 studies according to the first three principal components of their average GC-dependent sequencing efficiencies (Figs. S3 and S4), we were able to identify 4 qualitatively different shapes of GC-dependent efficiency curves (Fig. 4A). With an efficiency between 50% and 100% throughout the whole GC spectrum,

clusters I (10 studies) and II (16 studies) show a noticeable but limited effect of GC-content on sequencing efficiencies. In comparison, the studies in clusters III (3 studies) and IV (4 studies) exhibit a much stronger effect. Whereas in cluster III the sequencing efficiency is reduced only for GC-poor species, both GC-poor and GC-rich species are affected in cluster IV. Overall, we observe that GC-dependent sequencing efficiencies can vary drastically between otherwise similarly designed studies.

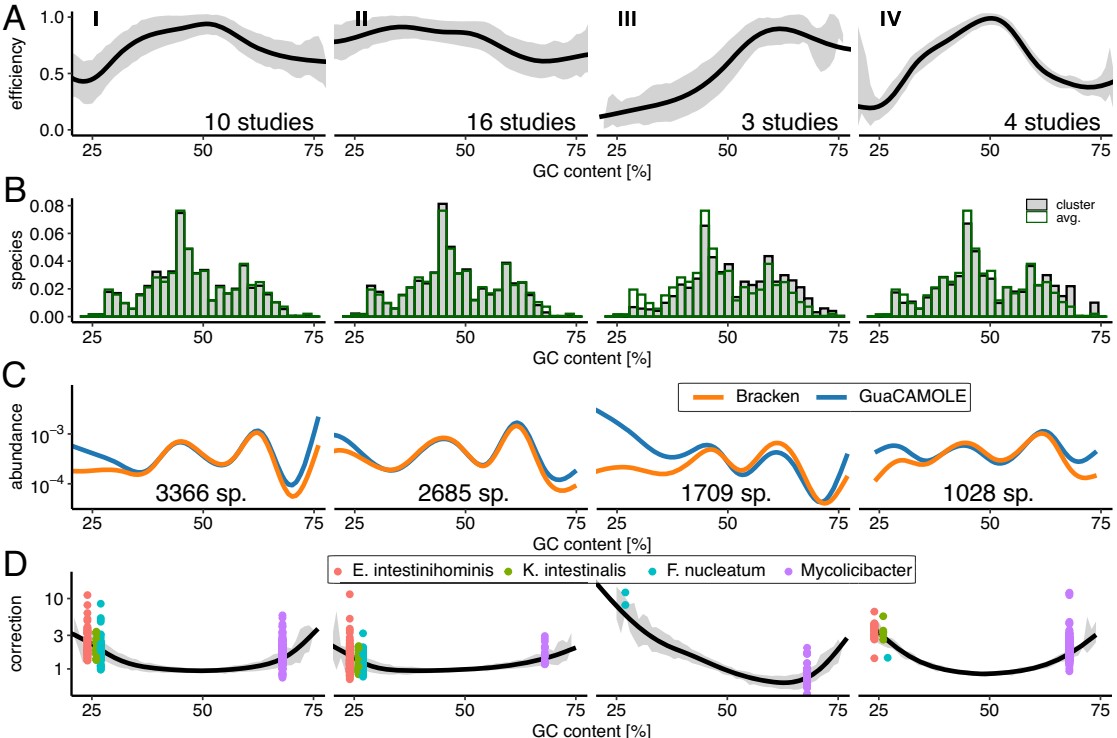

**Fig. 4 | Performance of GuaCAMOLE for human gut microbiomes.** GuaCAMOLE results for 3031 of 3435 samples from 33 studies of human gut microbiomes previously selected based on sample quality[27,28]. For the remaining 475 samples GuaCAMOLE was unable to reliably estimate efficiencies. Studies were clustered into 4 clusters based on the first 3 principal components of their average GC-dependent sequencing efficiency. In **A**, **C**, **D** lines are averages across the samples in a cluster smoothed with with ggplot's `stat_smooth`, method "GAM". The grey areas in panels **A**, **D** represent ± one standard deviation around the average. **A** GC-dependent sequencing efficiencies for each of the 4 clusters. **B** Distribution of genomic GC content of species detected in each cluster of studies. **C** Average species abundance as a function of GC content. **D** Abundance correction applied by GuaCAMOLE.

## Correct abundance estimation of GC-poor and GC-rich species

The distribution of average genomic GC content across detected species agrees between clusters I, II, and IV, but is noticeably shifted towards higher GC content for cluster III (Fig. 4B). This indicates that the loss of sequencing efficiency for GC-poor species for studies in cluster III is severe enough to cause species to be overlooked. For GC-rich species, the effect is reversed. Here, cluster III exhibits the highest sequencing efficiency, and as a result, more species with high GC content are detected in cluster III than in other clusters.

The abundances of species with differing genomic GC content show a similar trend (Fig. 4C, D). For GC-poor species in cluster III, the uncorrected estimates underestimate the abundance up to 10-fold. In clusters I and IV, both GC-rich and GC-poor species are underestimated up to 3-fold.

The taxa particularly strongly affected by GC-bias are *Endlipuvirus intestinihominis* (NCBI:txid2955861), *Kahucivirus intestinalis* (NCBI:txid2956048), *Fusobacterium nucleatum* (NCBI:txid851), and *Mycolicibacter* (NCBI:txid1073531). *F. nucleatum* in particular has been associated with colorectal cancer[18–20], and is consistently underestimated by Bracken due to GC-bias; on average 1.9-fold in cluster I (96 samples), 1.2-fold in cluster II (81 samples), 10-fold in cluster III (2 samples) and 1.4-fold in cluster IV (1 sample).

GC-bias affects not only the abundance estimates of individual species, but also summary statistics about the composition of microbial communities. In particular, we observe that for most samples the alpha diversity computed from uncorrected samples is noticeably lower than after GC-bias correction (Fig. S5).

## False positive removal

False positive taxa are created by reads that are wrongly assigned by Kraken to taxa not present in the sample. Filtering detected taxa based on read counts is a common way to remove some of these false positives, but this is not always effective[36]. GuaCAMOLE additionally filters taxa based on how well their observed reads match their reference genomes by comparing observed and expected GC distributions for each taxon (see Methods). Briefly, if observed and expected read counts across a taxon's GC bins vary more than a pre-defined threshold, the taxon is removed as an outlier, all abundances are recomputed, and another round of outlier detection is initiated (see Fig. S6 for an example). This process is repeated a user-defined number of times, by default 5. This default was set based on the mock community data of Tourlousse et al., where it reduces the number of false-positive taxa from 18 ± 9 to 8 ± 6 and only removes a true positive in two cases (protocols JH and B0). By adjusting these thresholds, users can trade sensitivity (i.e., detecting as many taxa as possible) against specificity (avoiding false positives) and accuracy of efficiency estimates (preventing false-positive taxa from interfering with the sequencing efficiency estimates).

For the simulated mock communities (Fig. 2B), GuaCAMOLE's outlier removal reduces the number of false-positive taxa from 342 ± 218 to 120 ± 86 (Fig. S7). Altogether, these taxa typically represent only a few percent of all reads (Fig. S8)). The more stringent filtering of GuaCAMOLE increases the false negatives (taxa which are present but not detected) from 6 ± 10 for Bracken to 13 ± 17 for GuaCAMOLE. MetaPhlAn4 is generally much more specific and less sensitive; it finds 35 ± 54 false-positive taxa but misses 54 ± 68 taxa actually present.

For users desiring maximal sensitivity and accuracy of abundance estimates without the risk of interference by false-positive taxa, GuaCAMOLE offers a mode that computes GC-corrected abundance estimates for all taxa detected by Bracken. In this mode, outlier removal affects only the set of taxa used to estimate GC-dependent efficiencies, and the estimated efficiencies are then used to correct the Bracken

abundance estimates for GC-bias (thus effectively borrowing information between taxa with similar GC content). This mode was used for the analysis of the CRC samples presented in Fig. 4.

## Discussion

GuaCAMOLE infers both bias-corrected abundances and GC-dependent sequencing efficiencies from a single sample without prior information about the amount or direction of GC-bias present in the data. The algorithm is agnostic about the specific sequencing protocol used and can correctly detect and correct for GC-bias without calibration data or prior knowledge about the expected type of bias. For most sequencing protocols, the bias-corrected abundances reported by GuaCAMOLE are more accurate than those reported by both Bracken and MetaPhlAn4. The advantage provided by GuaCAMOLE increases with the amount of GC bias present, and thus in particular with the amount of PCR amplification done prior to sequencing. Interestingly, we do not observe an advantage of MetaPhlAn4 over Bracken, even though we might expect the marker gene-based approach of MetaPhlAn4 to be less susceptible to bias. In fact, Bracken and MetaPhlan4 often show a relatively similar quantification error. This further corroborates that the improvement offered by GuaCAMOLE does indeed stem from successful correction of GC bias and not from other algorithmic differences.

In addition to bias-corrected abundances, GuaCAMOLE reports accurate GC-dependent sequencing efficiencies. This is useful as a quality control to check that library preparation and sequencing perform as expected. It also provides a way to estimate the amount of bias that affects taxa that remained unobserved. Finally, it allows different library preparation and sequencing protocols to be compared without the need for mock communities with known abundances.

GC bias can affect the abundance estimates of clinically relevant pathogens such as *F. nucleatum*, which has been associated with a range of diseases[18–20]. We observe this in published microbiomes of colorectal cancer patients, where, depending on the study, the abundance of *F. nucleatum* is often underestimated 2-fold, and can be underestimated up to 10-fold before GC bias correction. Generally, we observe that the under-estimation of GC-poor and GC-rich species can differ widely between different studies of human gut microbiomes. One cause for GC-bias is likely the commonly used Nextera XT library preparation kit[5,17]. However, the four qualitatively different shapes of GC-bias we observe in real-world human gut microbiome data suggest that many other common library preparation techniques introduce biases as well.

Even uniform biases can skew comparisons between different samples under some circumstances[21,37]. The large qualitative difference between the GC-biases affecting different real-world human gut microbiome studies thus poses a risk for meta-analyses such as refs. 27,28. While the quantitative effect of such will depend heavily on the design of such meta-analyses, GuaCAMOLE can help to mitigate these risks by uncovering the GC-bias affecting different studies and by offering a way to correct it.

GuaCAMOLE detects false-positive taxa by checking for outliers within the deviations of observed from expected read counts. This offers more power to detect false-positive taxa than read-count thresholding and ensures that such outliers do not skew the estimated sequencing efficiencies and abundances. However, this false-positive detection assumes reasonably accurate reference genomes. Therefore, taxa that are present but whose reference genomes are inaccurate are at risk of being flagged as false positives and subsequently removed. If this is a concern, the GC-dependent efficiencies reported by GuaCAMOLE can be used to correct the bias present in the abundances estimated by other tools such as Bracken or MetaPhlAn4. For Bracken, GuaCAMOLE already implements this mode of operation as an option.

GuaCAMOLE relies on the overlap between the GC distributions of taxa to estimate sequencing efficiencies. For small communities, particularly if most taxa have an extreme genomic GC content, insufficient overlap can reduce the accuracy of GuaCAMOLE. In such cases, the fraction of reads assigned to taxa flagged as false-positives is typically also high. As a safeguard, GuaCAMOLE thus warns the user and refuses to report estimates if that fraction exceeds 50%.

The runtime of GuaCAMOLE is mostly on par with most other tools, if slightly slower. The longer runtime is to a large degree caused by the need to re-read all sequencing reads to compute their GC content. While GuaCAMOLE is even now fast enough to be practical, a future optimization could be to modify Kraken2 to compute each read's GC content during taxonomic mapping. Doing so would remove the need for GuaCAMOLE to access the sequencing data and would improve performance considerably.

GuaCAMOLE provides a computational method to detect and correct for GC bias in sequencing protocols. For a wide range of sequencing protocols, GuaCAMOLE substantially improves abundance estimates over alternative methods. Taxa whose abundance estimates are improved include clinically relevant species. GuaCAMOLE is in principle applicable to all types of metagenomic samples, but performs best when reasonably complete reference genomes are available for all species. To facilitate its use by the community and its integration into standard pipelines, GuaCAMOLE is available as an easy-to-use and fast Python package under https://github.com/Cibiv/GuaCAMOLE.

## Methods

### The GuaCAMOLE algorithm

GuaCAMOLE operates on a pre-defined taxonomy comprising nodes $K_1, \ldots, K_n$, which are arranged in a tree. We often refer to these nodes simply as taxa. Leaf nodes represent individual species (or strains), whereas internal nodes represent higher taxonomic groups such as genera, families etc. Leaf nodes $K_j$ always have an associated genome $G_j$, for internal nodes, this is optional. The taxonomy, together with all associated genomes, is referred to as a database.

GuaCAMOLE estimates the abundances of these taxa from a metagenomic sequencing library containing a number $N$ of sequencing reads. Each read is assumed to stem from one of the taxa in the taxonomy. The GC content of a read is the fraction of bases that are either guanine (G) or cytosine (C). We assign reads into one of $b$ equally spaced bins according to their GC content, and denote the GC content by the index $g$ of the respective bin.

GuaCAMOLE assumes that the composition of the sequencing library depends on (i) the abundances $a_1, \ldots, a_n$ of all taxa (zero for all taxa not present in the sample), (ii) the GC-dependent sequencing efficiency $\eta_g$, (iii) the genomic GC content distributions $f(j, g)$ of the taxa (defined as the expected fraction of reads from taxon $j$ which fall into GC bin $g$, normalized such that $\sum_g f(j, g) = 1$; see section *the genomic GC content distributions* below), and (iv) the lengths $l_1, \ldots, l_n$ of the taxa's genomes. In terms of these quantities, GuaCAMOLE assumes that the number $O(j, g)$ of fragments stemming from taxon $j$ with GC content $g$ in the library is

$$O(j,g) = N \cdot \frac{a_j \cdot \eta_g \cdot l_j \cdot f(j,g)}{\sum_{g=1}^{b} \sum_{i=1}^{n} a_i \cdot \eta_g \cdot l_i \cdot f(i,g)}. \tag{1}$$

Note that abundances $a_i$ and efficiencies $\eta_g$ are defined only up to a factor by Eq. (1), we normalize these quantities by demanding that $\sum_j a_j = \max_g \eta_g = 1$.

GuaCAMOLE estimates abundances and GC-dependent efficiencies by plugging observed fragment counts $O(j, g)$, GC distributions $f(j, g)$, and genome lengths $l_g$ into Eq. (1) and solving for $a_1, \ldots, a_n$ and $\eta_1, \ldots, \eta_b$. Note that this system is typically over-determined: it contains on the order of $nb$ equations for $n + b$ unknowns.

## The number of reads per taxon and GC bin

To compute observed read counts $O(j, g)$, reads are first assigned to taxonomic nodes using Kraken2[8]. This yields the number of assigned reads (read pairs for paired-end libraries) $M(j)$ for every node $j$ in the taxonomy. The reads assigned to each node are then further subdivided according to their GC content to obtain $M(j, g)$, the number of reads assigned to taxon $j$ with GC content $g$.

The counts $M(j, g)$, however, are biased by similarities between genomes of different taxa. When Kraken2 is unable to unambiguously assign a read to a taxon due to such similarities, Kraken2 assigns those reads to the lowest common ancestor (LCA) of all matching taxa. To correct for this systematic bias, we use the same approach as the Bracken algorithm introduced by Lu et al. [29]. We recall that $G_j$ denotes the genome associated with node $K_j$ in the taxonomy, which always exists for leaf nodes but is optional for internal nodes. Like Bracken, we compute the conditional probabilities $P(r \in G_j \mid K_i)$ that a read which was assigned to taxon $K_i$ actually stems from a descendant $K_j$ of $K_i$ with associated genome $G_j$ in the taxonomic tree (see "The GuaCAMOLE algorithm"). This is done by first computing $P(K_i \mid r \in G_j)$, the probability that a read stemming from genome $G_j$ is assigned to taxon $i$, by finding the taxon assigned to every possible read from genome $G_j$ (of the same length as in the data). The desired probabilities $P(r \in G_j \mid K_i)$ are then found by applying Bayes' theorem, see ref. [29] for details. The original Bracken algorithm uses $P(r \in G_j \mid K_i)$ to redistribute reads assigned to $K_i$. GuaCAMOLE follows the same approach, but additionally keeps track of the GC content when redistributing reads. To estimate the number of reads in GC bin $g$ that stem from taxon $j$, we thus compute

$$\tilde{O}(j, g) = \sum_i M(i, g) \cdot P(r \in G_j | K_i) \tag{2}$$

where the sum runs over all ancestors $K_i$ of $K_j$ in the taxonomic tree (if $K_i$ is not an ancestor of $K_j$, $P(r \in G_j \mid K_i) = 0$). We emphasize that the total number of reads is invariant under the redistribution done by Eq. (2); this is ensured by the fact that $\sum_j P(r \in G_j \mid K_i) = 1$. In particular, reads redistributed from $K_i$ to a descendant $K_j$ are removed from $K_i$. We also note that we have assumed here for simplicity that $P(r \in G_j \mid K_i)$ does not depend on the read's GC content.

To make it possible to compare abundances of higher taxonomic levels, such as genera, we then sum the corrected read counts over descendants. The per-taxon and per-GC-bin read counts plugged into Eq. (1) are thus

$$O(j, g) = \tilde{O}(j, g) + \sum_{i \in D_j} \tilde{O}(i, g) \tag{3}$$

where $D_j$ denotes the descendants of node $K_j$.

## The genomic GC content distributions

For Eq. (1) to hold, the observed counts $O(j, g)$ must, in theory, arise by sampling from the genomic GC content distributions $f(j, g)$. These distributions must thus take the redistribution of reads into account. To find $f(j, g)$, we first compute the individual GC distributions $q(j, g)$ of the genomes in the taxonomy. Given the fragment length $\ell_f$ and read length $\ell_r$ of the experimental data, $q(j, g)$ reflects the fraction of windows of length $\ell_f$ whose GC content within the parts covered by reads (i.e. the first and last $\ell_r$ bases for paired-end reads) is $g$. Here, we use the correct experimental fragment- and read length to avoid systematic errors. We then find the expected GC content distribution of the reads assigned to a specific taxon

$$h(j, g) = \sum_{i \in D_j} q(i, g) \cdot P(r \in G_i | K_j) \tag{4}$$

where the sum runs over the descendants of $K_j$ (otherwise, $P(r \in G_i \mid K_j) = 0$). In Eq. (4), we have thus propagated the GC distributions of individual genomes *upwards* in the taxonomic tree to account for the fact that Kraken2 will assign some reads to taxa at higher taxonomic levels. We now propagate these mixed GC distributions of internal nodes back *downwards* to find the expected GC distribution after fragment redistribution. Mimicking Eq. (2) we thus compute

$$\tilde{f}(j, g) = \frac{\sum_i h(i, g) \cdot M(i) \cdot P(r \in G_j | K_i)}{\sum_i M(i) \cdot P(r \in G_j | K_i)} \tag{5}$$

where the sums run over the ancestors of taxon $K_j$. Finally, we proceed similarly to Eq. (3) and average the GC distributions of all descendants, weighted by their fragment counts,

$$f(j, g) = \frac{\tilde{f}(j, g) \cdot \tilde{O}(j, g) + \sum_{i \in D_j} \tilde{f}(i, g) \cdot \tilde{O}(i, g)}{\sum_{\gamma=1}^b \left( \tilde{f}(j, \gamma) \cdot \tilde{O}(j, \gamma) + \sum_{i \in D_j} \tilde{f}(i, \gamma) \cdot \tilde{O}(i, \gamma) \right)}. \tag{6}$$

Since the tails of these distributions are typically noisy, we restrict these distributions to the range between the 2.5% and 97.5% quantiles for every taxon $j$.

## Genome lengths

We assign a single genome length $\ell_j$ to every taxon $j$, independent of its taxonomic level or number of associated genomes. To do so, we average over the lengths of all assigned genomes of a taxon and all of its descendants. To account for the observed read distribution, we weight each genome length with the prior probability $P(K_j)$ that a random read stems from taxon $j$ as computed by Bracken[29].

## Estimating abundances

To estimate abundances $a_1, \ldots, a_n$, we rearrange Eq. (1) into the following expression for the GC-dependent efficiencies $\eta_g$ in bin $g$,

$$\eta_g = \frac{C}{N} \cdot \underbrace{\frac{O(j, g)}{l_j f(j, g)}}_{\text{Obs/Exp reads}} \cdot \frac{1}{a_j}. \tag{7}$$

where $C = \sum_{g=1}^b \sum_{i=1}^h a_i \cdot \eta_g \cdot l_i \cdot f(i, g)$ is a normalization factor. Note the correspondence to Fig. 1: For a fixed taxon $j$, $\eta_g$ is proportional to the obs/exp ratio $O(j, g)/l_j f(j, g)$. After scaling with inverse abundances, $a_j^{-1}$ these ratios become comparable across taxa.

Given abundances $a_1, \ldots, a_n$, Eq. (7) provides a separate estimate of $\eta_g$ for every taxon whose genomic GC distribution overlaps $g$. This allows us to estimate the abundances by maximizing the agreement between these separate estimates of $\eta_g$. In terms of the inverse abundances, $a_1^{-1}, \ldots, a_n^{-1}$ this can be expressed as the minimization of the quadratic form

$$G(a_1^{-1}, \ldots, a_n^{-1}) = \sum_{i=1}^n \sum_{\substack{j=1 \\ i \neq j}}^n \sum_{g=1}^b \left( \frac{O(i, g)}{l_i f(i, g)} \cdot a_i^{-1} - \frac{O(j, g)}{l_j f(j, g)} \cdot a_j^{-1} \right)^2. \tag{8}$$

Here, we have dropped the pre-factor $C/N$ from the GC-dependent efficiencies $\eta_g$. In practice, we drop all terms from the sum in Eq. (8) that are either undefined or unreliable. A term is undefined if one of the two GC distributions does not overlap bin $g$ (i.e., $f(i, g)$ or $f(j, g)$ is undefined). A term is considered to be unreliable if the total number of reads assigned to one of the taxa including descendants (i.e. $\sum_{D_j} M(j)$, for taxon $j$, similarly for $i$) lies below some user-defined threshold (minimum read threshold, default 500).

**Regularization.** If the taxa in a sample can be partitioned into two sets $A$ and $B$ such that Eq. (8) contains no term containing an abundance from $A$ and an abundance from $B$, the relative abundances between sets $A$ and $B$ are undefined. In Fig. 1, this would be represented as two groups of taxa whose GC efficiency curves do not mutually overlap. In this situation, Eq. (8), as stated, does not have a unique minimum. To avoid this, we regularize the quadratic form $G$ by penalizing large differences in efficiency between neighboring GC bins. Using Eq. (7), we express $\eta_g$ sans the prefactor $C/N$ as a weighted average of taxon-specified efficiencies,

$$\lambda_g = \frac{1}{n_g} \sum_{i=1}^{n} \log(O(i,g)+1) \cdot \frac{O(i,g)}{l_i f(i,g)} \cdot a_i^{-1}. \qquad (9)$$

We now define the regularized objective function

$$\tilde{G}(a_1^{-1}, \cdots, a_n^{-1}) = \frac{1-r}{n^2} G(a_1^{-1}, \cdots, a_n^{-1}) + r \sum_{k=1}^{b} \sum_{l=1}^{b} e^{-|k-l|} (\lambda_k - \lambda_l)^2 \qquad (10)$$

which is still quadratic since $\lambda_g$ is linear in $a_1^{-1}, \ldots, a_n^{-1}$. The regularized program is thus still efficiently solvable. Here, $r$ is a hyperparameter that controls the amount of regularization to apply. Smaller values of $r$ allow more extreme and small-scale variations in sequencing efficiency to be corrected, but increase the risk of incorrect estimates in the case of taxa partitions with non-overlapping GC distributions.

To find abundances $a_1, \ldots, a_n$, we first minimize the regularized objective function $\tilde{G}$ in terms of $a_1^{-1}, \ldots, a_n^{-1}$ subject to $\sum_i a_1^{-1} = 1$ using the Python package *cvxopt*. We then compute $a_1, \ldots, a_n$, normalized such that $\sum_j a_j = 1$, and compute the GC-dependent sequencing efficiencies $\eta_g = \lambda_g / \max_\gamma \eta_\gamma$.

### False-positive detection and removal

The set of taxa with a non-zero number $O(j, g)$ of assigned reads often contains taxa that are not actually present in the sample. The reads assigned to such a false-positive taxon are consequently not uniformly random draws from the taxon's genome, and we hence expect to see some deviation from Eq. (1). To detect false-positives we therefore look for outliers amongst the relative residuals $\xi(j,g) = (O(j,g) - \bar{O}(j,g))/\bar{O}(j,g)$, where $\bar{O}(j,g)$ is the expected number of reads computed using Eq. (1). Taxa are removed if the variation $\phi(j) = \max_g \xi(j,g) - \min_g \xi(j,g)$ of $\xi(j, g)$ of their residuals $\xi(j, g)$ exceeds a predefined threshold $T$. After removing taxa, all abundances are re-computed, the threshold is halved, and another round of false-positive removals is done. We stop after a specified number of rounds (per default 5).

### Simulated mock community data

We generated synthetic communities by sampling 5, 10, 50, 100 or 400 bacterial genome assemblies from RefSeq. Only assemblies of taxa that are represented with at least one genome in our Kraken2 database were considered, but we allowed the strains to differ. To control the distribution of genomic GC content within generated communities, assemblies were partitioned into $N = 12$ equally sized GC-bins according to their genomic GC content and sampled in a two-stage process. First, a GC-bin $i \in \{1, \ldots, 12\}$ was drawn either with (i) uniform probability $P_i^{unif} = 1/N$, (ii) with probabilities $P_i^{med} \propto \exp(-(g-N/2)^2 18/N^2)$ skewed towards a GC content of 50%, (iii) with probabilities $P_i^{ext} \propto 1 - P_i^{med}/\max_j P_j^{med}$ skewed towards extreme GC content. From the selected GC bin, an assembly was then drawn randomly. After sampling the genomes present in each community, the corresponding species abundances were sampled from a log-normal distribution. Finally, a paired-end sequencing library was generated for

each community using a modified version of InSilicoSeq v2.0.1[38], available under https://github.com/Cibiv/InSilicoSeq-GCBias. In our modified version of InSilicoSeq, a bias against reads with extremal GC content is introduced by rejecting reads with probability $10 \cdot (g-0.5)^2$.

### Experimental mock community data

For the analysis of the mock community data of Tourlousse et al. [16] (SRA accession SRS7661134), we used the RefSeq release 220 database containing human, archaeal, viral, plasmid, and bacterial DNA[39]. We ran GuaCAMOLE (in taxonomic abundance mode), Bracken, and MetaPhlAn4. For Bracken and GuaCAMOLE, we set the read threshold to 500. For GuacAMOLE we set the number of false-positive removal rounds to 5 (4 for JH and B0), the read length to 150bp and the fragment length to the value observed for each protocol: 200bp for D0, IL and IH, 250bp for DH, BL, BH, EH, EL, 325bp for F0, FL, FH, 350bp for JH, AH, C0, CL, 400bp for AL, A0, JL, and 300bp for all other protocols. For MetaPhlAn4, we use the CHOCOPhlAn database v202103 with default parameters, and manually corrected the classification of *F. prausnitzii* to *F. duncaniae* since this recent reclassification is not reflected by CHOCOPhlAn v202103. Since Bracken always reports sequence abundances, we adjusted the Bracken-estimated abundances using the same genome length estimates that GuaCAMOLE uses to make them comparable. For Sylph, we profiled the reads using the GTDB database v220[40]. To ensure comparability across all methods, we manually mapped the taxonomic profiles from Sylph, mOTUS3, and SingleM to their corresponding NCBI taxonomy IDs using the provided genus and species names.

### Analyzing colorectal cancer microbiome data

We ran GuaCAMOLE on all paired-end human gut microbiome samples with non-zero read length from the curated list of samples from 33 studies found in Table S2 of Murovec et al. [28]. For the study of Yachida et al. [26] we included all 645 samples, even those not included in the list of Murovec et al. GuaCAMOLE was run with default parameters except for activating genome length correction and specifying the read length reported in the SRA metadata of each sample. We then averaged the inferred GC-dependent sequencing efficiencies across each of the 33 studies (Fig. S3), performed a principal component analysis on the resulting efficiency curves, and clustered the studies based on the first three principal components using hierarchical clustering (R's hclust command) with Euclidean distances (Fig. S4). Based on visual inspection of the resulting dendrogram, we split the studies into 4 clusters. For each cluster, we computed the number of species, average corrected abundance, and average abundance correction compared to Bracken for each GC-bin from 25% to 75% (Fig. 4). As "corrected abundances" we used the abundances corrected based on the inferred sequencing efficiency of each taxon and we included taxa flagged by GuaCAMOLE's false-positive detection. For plotting, abundances and abundance corrections were smoothed with R's stat_smooth command with method 'GAM'.

### Reporting summary

Further information on research design is available in the Nature Portfolio Reporting Summary linked to this article.

## Data availability

The sequencing data used in this study is publicly available from the short read archive (SRA), the data of Tourlousse et al. [16] under accession PRJNA650228 and the data of Mori et al. [35] under accession PRJNA650228. The SRA accessions of all samples used in this study, including the curated colorectal cancer (CRC) samples from refs. 27,28, together with the processed data and scripts required to reproduce the main analyses and figures of this publication are available at https://github.com/Cibiv/GenomicGCBiasCorrectionImproves AbundanceEstimation.

## Code availability

GuaCAMOLE is available at https://github.com/CIBIV/GuaCAMOLE under an open-source license. All analyses in this study were done using with version ref. 41.

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

## Acknowledgements

We thank Simon Haendeler and Fausto Bradke for helpful suggestions regarding algorithm implementation and optimization. For help and

support with the high-performance computing (HPC) infrastructure we thank Robert Happel and the Scientific Computing and Data Analysis (SCDA) section of Core Facilities at OIST. Finally we thank all members of the Biological Complexity Unit at OIST, the Center for Integrative Bioinformatics Vienna (CIBIV) and the Vienna Biocenter PhD program for inspiring discussions.

## Author contributions

L.H., F.G.P. and Av.H. designed the algorithm; L.H. implemented, tested and benchmarked the algorithm; F.G.P. and L.H. applied the algorithm to human gut microbiomes; L.H. and F.G.P. created the figures; L.H., Av.H., and F.G.P. drafted and revised the manuscript; Av.H. and F.G.P. supervised the project.

## Competing interests

The authors declare no competing interests.
