## [Transparent Peer Review file · Nature Communications]

Genomic GC bias correction improves species abundance estimation from metagenomic data

Corresponding Author: Dr Florian Pflug

Version 0:

Reviewer comments:

Reviewer #1

(Remarks to the Author)

The study includes a novel computational method (GuaCAMOLE) which detects and removes GC bias from metagenomic sequencing data. To demonstrate their approach, the authors analyzed experimental mock community data to confirm both estimates to be accurate across a wide range of experimental protocols. They also show based on their tool that F. nucleatum which has been associated with colorectal cancer is consistently underestimated by Bracken in cancer samples. The tool seems helpful, but its performances are not fully clear to me (see suggestions below) and thus it is not clear to me if the paper fits Nature communication or to a journal with lower IF.

Major

1) GC content is correlated in many cases with gene features (e.g. functionality and expression) and phenotypes of organisms. In addition, when we analyze metagenomic sequencing we usually do not know the composition of the organisms in the sample. Thus, to understand the new methods the authors should generate various examples in-silico samples of metagenomic data and analyze it with their tool in comparison to other tools. The in-silico set of examples should include various numbers of organisms (e.g. between 5 to 400), different ratios of organismal abundance (e.g. an organism can be very rare: 0.001 % and up to very abundance: 10%), the GC content of the organisms in the community can be similar (including very high or very low) or extremely non-similar (very high GC and very low GC). The genomes should be real known genomes and various types of errors and biases should be added (as in real sequencing). When simulating the analysis of the data with the tool you should assume that you do not know the composition of genomes in the data. To evaluate the performance of your tool and other tools you should compare the estimations to the actual inputs.

2) There are dozens of additional studies related to GC content correction that should be cited and additional tools that you should compare to your tool. A very partial list (see also the reference of these papers and the papers cite them):

[https://www.cell.com/cell/fulltext/S0092-8674\(19\)30775-5](https://www.cell.com/cell/fulltext/S0092-8674(19)30775-5)

<https://link.springer.com/article/10.1186/S40168-018-0541-1>

<https://genome.cshlp.org/content/26/12/1721.short>

<https://link.springer.com/article/10.1186/1471-2164-12-444>

3) "... Highlighted protocols GH, DH, IH, FH, IL.." I can not find where are these protocols defined? please make sure that all notation are defined.

(Remarks on code availability)

Reviewer #2

(Remarks to the Author)

The authors present GuaCAMOLE for discovery and removal of GC bias when profiling metagenomics samples. The method runs per a provided dataset using Kraken2 and Bracken helper tools and produces reference-free, protocol-specific corrected dataset abundances (a_i) alongside with the estimates of the protocol efficiency (η_g for each bin g of GC-content distribution for each taxon in the taxonomy). The tool is benchmarked using an existing mock community dataset of 20 taxa (19 species) sequenced using 28 different protocols showing better performance compared to Bracken and MetaPhlan4. The method is also applied to 80 gut microbiomes from a colorectal cancer datasets showing that low GC species could be underestimated in this dataset as well. Finally, the method is applied to outlier detection and removal reducing the number of false positive identifications. The source code is provided and the API is documented.

GC-bias and correction is a well-studied problem within sequencing and applied across many fields (RNA-seq, isolate sequencing, etc). Curiously not much have been published on GC correction in metagenomics (although some work exists). We find the work to be interesting, there are some additional benchmarks that will help highlight the method performance and a few points that require further clarifications.

Major comments:

GuaCAMOLE should be compared to more complex simulated or mock metagenomics datasets. Currently it is applied to a dataset of 20 genomes of which 19 are distinct species. One could use CAMI2 datasets and the CAMI2 profiling challenge, for instance the Marine dataset which also harbours a lot of difficult/unknown species. Additionally, two datasets from Zymo are available with ground truth abundances that would be beneficial to benchmark <https://zymoresearch.eu/collections/zymbiomics-microbial-community-standards/products/zymbiomics-gut-microbiome-standard> and <https://zymoresearch.eu/collections/zymbiomics-microbial-community-standards/products/zymbiomics-microbial-community-standard>.

There exists a plethora of taxonomic profilers for metagenomics data. Only Kraken2 and MetaPhlan are benchmarked at Figure 2B and Figure 3. It would help to highlight the benefits of the method by comparing to Sylph also based on k-mer search (Shaw and Yu, Nature Biotechnology, 2024) and SingleM (Woodcroft et al., biorxiv, 2024).

Currently, the presentation of the benefit of running GuaCAMOLE is not convincing and more work is needed to show the impact of using GuaCAMOLE. The method is applied to a colorectal cancer dataset where they show that the abundance for some species is changed. However what is the impact of this in the interpretation of the results? Does the conclusions or downstream predictions based on colorectal cancer change when Guacamole is used?

Second, why were only 80 samples analysed from Yachida et al? The study sequenced 616 samples. Furthermore, many more CRC microbiome datasets are available (<https://doi.org/10.1038/s41591-019-0405-7>, <https://doi.org/10.3389/fmicb.2024.1426407>). Could these be analysed as well? Is the correction of species consistent across the datasets?

While the method is created for evaluating abundances in a metagenomics dataset, it should be possible to apply the method to a "single-genome" dataset and evaluate the efficiency of the sequencing protocol (assuming the known abundance to be 1). Would the results reflect empirically evaluated protocol efficiencies as e.g. calculated here, Figure 2 <https://academic.oup.com/gigascience/article/9/2/giaa008/5735313>?

False positive identification: What was the abundance of the species that were false positives? I.e. were they low or high? One would expect that this was low abundance species, but interesting (and more important) if also higher level FP taxa were removed.

On the mock community data the authors restrict themselves to 3 rounds of outlier removal. Was this guided by knowing the ground truth? I.e. the hyperparameter that gave the best value? If so it should be noted (i.e. it can be overfitting when train/test is the same).

What is the run time and resourced required? This should also be compared against the benchmarked methods.

In the section 6.2, the Eq (2) uses the term $P(r \in G_j | K_i)$. Even though the reference to Bracken is in place, it is unclear from the text how such probability is computed. Since it's an important part of the method, briefly describing in the text is necessary. Also, the important notations G_j and K_i are not explained. Does $P(r \in G_j | K_i)$ only apply to leaf descendants, all descendants or all descendants that have associated genomes?

Eq (2) makes a summation over the whole taxonomy tree, while it seems that the non-zero probabilities can only exist for the ancestors of the node j . If that is the case, rewriting it in the terms of ancestors would improve the readability.

The calculations in Eq (3) are done recursively. Is it possible for the non-leaf node j to have $_O(j,g)$ that is not equal to zero? If yes, can it lead to double counting of the number of the assigned reads?

In Eq (5), the term $h(j,g)$ that already aggregates over all the nodes multiplied by $P(r \in G_j | K_i)$ is used to again aggregate and multiply by the same term over all the nodes. Does it lead to double accounting of $P(r \in G_j | K_i)$, thus leading to overestimation of its effect?

The authors provide the test data in the repository to perform an analysis on real data. While it is an important and valuable test, it would help both the code quality and the method clarity if the repository would also have a test case with mock data. We suggest creating a mock taxonomy with 5-10 nodes, and abstract the variables that come from third-party tools, such as $P(r \in G_j | K_i)$ to have mock values. In this case, a more clear interface between the novel method and the third-party software will be enforced and the testability and the clarity of the code will improve. Right now in order to test if the software works, one has to first build Kraken and Bracken databases. It should be possible to run a unit test without them.

Minor comments:

“Read counts in each taxon-GC-bin are then normalized...” - the concept of taxon-GC-bin was not introduced before, making it hard to follow the logic of the method.

“(iii) the genomic GC content distributions...” - j was not previously defined.

Are the simulated reads from the section 6.3 also assigned to a specific taxon by running Kraken2 or the ground truth values are used?

“Finally, we proceed similarly to Eq. (3) and average the GC distributions of all descendants, weighted by their fragment counts...” - in the Eq (6), there is an inconsistent use of γ in both numerator and denominator. In numerator, it seems like there is a typo under the nested summation, and γ is meant to be g . In denominator, γ is appearing in the two nested summation terms. Probably, the use of γ' should be preferred for the nested summation term.

The lines in the provided PDF are not numbered which makes it slightly more inconvenient to comment on a specific place in text.

(Remarks on code availability)

Reviewer #3

(Remarks to the Author)

(Remarks on code availability)

Version 1:

Reviewer comments:

Reviewer #1

(Remarks to the Author)

The paper was improved but there are still open questions:

- 1) Why in figure 2 we do not see all the tools that appear in figure 3? You should add comparison to all of them to figure 3.
- 2) It is hard to re-review the paper without marking the new changes. You should provide a version with track changes relatively to the first version.
- 3) Need to better discuss in the discussion section the cases that the suggested approach fails (e.g. for very small communities whose taxa all have extreme GC content). Is there a way to fix this ?
- 4) Need to provide in the supplementary the raw data used to generate the figures (e.g. figure 2 – what are the genomes used)
- 5) In figure 2B the plots of the 3 algorithms are not on the same x-axis value: is this due to visualization reasons (in this case, it should be explained in the caption) or is it a real phase in the number of taxa (in this case, it should be fixed).
- 6) I expect to see more points in each group in 2B.

(Remarks on code availability)

Reviewer #2

(Remarks to the Author)

The authors have greatly improved the manuscript and we have no further comments

(Remarks on code availability)

Reviewer #3

(Remarks to the Author)

(Remarks on code availability)

Dear Editor and Reviewers,

Based on the detailed reviews, we have extensively updated and improved our manuscript. In addition to a larger number of new simulation studies, we have greatly expanded the number of real-world colo-rectal cancer (CRC) datasets from 80 to over 3,000. The analysis of the GC-bias present in those samples has uncovered that real-world CRC data exhibit large differences in the amount and type of GC-bias present. Since such large differences in GC-bias potentially affects meta-studies, we believe this additional result to be of great interest to the community.

Obtaining the necessary computational resources to download, store and analyze these additional datasets unfortunately took longer than initially expected; we apologize for the resulting delay in resubmitting our manuscript.

We thank the Reviewers for their many helpful comments and suggestions. In particular, we thank Reviewers 2 & 3 for suggesting the references which provided us with a curated list of publicly available CRC samples.

Sincerely,

Florian Pflug
(on behalf of all co-authors)

Reviewer 1

*The study includes a novel computational method (GuaCAMOLE) which detects and removes GC bias from metagenomic sequencing data. To demonstrate their approach, the authors analyzed experimental mock community data to confirm both estimates to be accurate across a wide range of experimental protocols. They also show based on their tool that *F. nucleatum* which has been associated with colorectal cancer is consistently underestimated by Bracken in cancer samples. The tool seems helpful, but its performances are not fully clear to me (see suggestions below) and thus it is not clear to me if the paper fits Nature communication or to a journal with lower IF.*

Major comments

1) GC content is correlated in many cases with gene features (e.g. functionality and expression) and phenotypes of organisms. In addition, we when we analyze metagenomic sequencing we usually do not know the composition of the organisms in the sample. Thus, to understand the new methods the authors should generate various examples in-silico samples of metagenomic data and analysis it with their tool in comparison to other tools. The in-silico set of examples should include various numbers of organisms (e.g. between 5 to 400), different ratios of organismal abundance (e.g. an organism can be very rare: 0.001 % and up to very abundance: 10%), the GC content of the organisms in the community can be similar (including very high or very low) or extremely non-similar (very high GC and very low GC). The genomes should be real known genomes and various types of errors and biases should be added (as in real sequencing). When simulating the analysis of the data with the tool you should assume that you do

not know the composition of genomes in the data. To evaluate the performance of your tool and other tools you should compare the estimations to the actual inputs.

We have updated Fig. 2 to include simulations of the kind suggested by the reviewer. Briefly, we simulated metagenomic libraries comprising from 5 to 400 species, selected randomly from RefSeq based on their genomic GC content according to one of three GC profiles: Predominantly species with extreme GC content (extreme), predominantly species with GC content around 50% (medium), and uniformly distributed GC content (uniform). In the analysis, we used the full RefSeq database. We compare GuaCAMOLE against the other RefSeq-based algorithms. Algorithms not based on RefSeq were omitted due to issued with unambiguous mapping of species names between databases.

2) There are dozens of additional studies related to GC content correction that should be cited and additional tools that you should compare to your tool. A very partial list (see also the reference of these papers and the papers cite them):

*[https://www.cell.com/cell/fulltext/S0092-8674\(19\)30775-5](https://www.cell.com/cell/fulltext/S0092-8674(19)30775-5)
<https://link.springer.com/article/10.1186/S40168-018-0541-1>
<https://genome.cshlp.org/content/26/12/1721.short>
<https://link.springer.com/article/10.1186/1471-2164-12-444>*

We have added additional citations where relevant, and added an explanation that we limit our discussion to the case of known reference genomes, which excludes methods such as MetaWRAP that deal with meta-genomic assembly. We have also added additional algorithms (Sylph, mOTUS and SingleM) to the comparison in Fig. 3, so that we now compare six algorithms in total (GuaCAMOLE, Bracken, MetaPhlAn plus the three newly added ones). While the average errors across all samples differ between algorithms, we find that all algorithms except GuaCAMOLE show similar amounts of GC bias in their abundance estimates (Figs. 3D and 3E). We therefore believe that this selection of algorithms is quite representative for algorithms which do not explicitly correct for GC-bias in their estimates.

3) "... Highlighted protocols GH, DH, IH, FH, IL..." I can not find where are these protocols defined? please make sure that all notation are defined.

We have added additional references to Table 1 where these abbreviations are defined.

Reviewers 2 & 3

The authors present GuaCAMOLE for discovery and removal of GC bias when profiling metagenomics samples. The method runs per a provided dataset using Kraken2 and Bracken helper tools and produces reference-free, protocol-specific corrected dataset abundances (a_i) alongside with the estimates of the protocol efficiency (η_g for each bin g of GC-content distribution for each taxon in the taxonomy). The tool is benchmarked using an existing mock community dataset of 20 taxa (19 species) sequenced using 28 different protocols showing better

performance compared to Bracken and MetaPhlan4. The method is also applied to 80 gut microbiomes from a colorectal cancer datasets showing that low GC species could be underestimated in this dataset as well. Finally, the method is applied to outlier detection and removal reducing the number of false positive identifications. The source code is provided and the API is documented.

GC-bias and correction is a well-studied problem within sequencing and applied across many fields (RNA-seq, isolate sequencing, etc). Curiously not much have been published on GC correction in metagenomics (although some work exists). We find the work to be interesting, there are some additional benchmarks that will help highlight the method performance and a few points that require further clarifications.

Major comments

1. GuaCAMOLE should be compared to more complex simulated or mock metagenomics datasets. Currently it is applied to a dataset of 20 genomes of which 19 are distinct species. One could use CAMI2 datasets and the CAMI2 profiling challenge, for instance the Marine dataset which also harbours a lot of difficult/unknown species. Additionally, two datasets from Zymo are available with ground truth abundances that would be beneficial to benchmark

We have added a comparison of GuaCAMOLE to Bracken and MetaPhlan4 for complex simulated communities ranging from 5 to 400 taxa with varying GC content distributions across taxa (see updated Fig. 2). For these simulations, taxa were drawn randomly from RefSeq, and the full RefSeq database was used during analysis (see Methods for details). We compared only RefSeq-based algorithms since (i) the simulations are based on RefSeq which disadvantages non-RefSeq-based algorithms and (ii) ambiguous species mappings between databases make meaningful comparisons difficult.

We also added an analysis of experimental data for a mock community designed to represent the human gut (Fig. A1). Here, we again used the whole RefSeq database in our analysis to create realistic conditions. Since both the the origin publication (Mori et al., 2023. DNA Research, **30-3** 30, 10.1093/dnares/dsad010.) and our analysis shows that taxon abundances differ significantly from the design goal of equal number of cells, we restricted ourself to a comparison of Bracken and GuaCAMOLE to ascertain the magnitude of GuaCAMOLE's corrections. Quite notably, we find that the choice of sequencing platform has a larger effect on GC-bias than the choice of DNA extraction protocol. Both these analyses by and large confirm the results seen previously.

We have attempted to run GuaCAMOLE on some of the CAMI2 challenges, but found that miss-match between the GC distributions of the sequences present in these samples and the reference genomes of the corresponding species is too large for GuaCAMOLE to produce meaningful results. In most cases, more than 50% of the reads are assigned to species which are later classified as false-positives. As we mention in the discussion, GuaCAMOLE requires reasonably accurate reference genomes to function; CAMI2 on the other hand seems to have been specifically designed to include difficult/unknown species.

2. There exists a plethora of taxonomic profilers for metagenomics data. Only Kraken2 and MetaPhlan are benchmarked at Figure 2B and Figure 3. It would

help to highlight the benefits of the method by comparing to Sylph also based on k-mer search (Shaw and Yu, *Nature Biotechnology*, 2024) and SingleM (Woodcroft et al., *bioRxiv*, 2024).

We have added all suggested taxonomic profiles to the comparison in Fig. 3. We did not add these additional algorithms to the comparison in Fig. 2A since that figure is intended only to show that GuaCAMOLE works as designed and does indeed recover different types of GC bias. In Fig. 2B, we restricted the analysis to RefSeq-based algorithms to ensure that mapping ambiguities of taxa between different databases do not skew the results.

3. Currently, the presentation of the benefit of running GuaCAMOLE is not convincing and more work is needed to show the impact of using GuaCAMOLE. The method is applied to a colorectal cancer dataset where they show that the abundance for some species is changed. However what is the impact of this in the interpretation of the results? Does the conclusions or downstream predictions based on colorectal cancer change when Guacamole is used?

We have greatly expanded the number of CRC samples we analyze with GuaCAMOLE. One of our new results is that the shape of GC-bias differs significantly even between different studies of human gut microbiomes (Fig. 4). We therefore find this large variation in GC-bias noteworthy, and slightly concerning. Even uniform biases have been shown to skew sample comparisons in some circumstances (McLaren, 2019; McLaren, 2024). It therefore stands to reason that meta-analyses such as those in Refs. (Gupta et al., 2020) and (Murovec et al., 2024) are very likely affected by significant and non-uniform GC-bias in the studies they use.

The magnitude of such an effect typically depends very strongly on the circumstances such as whether individual studies are balanced (contain the same number of healthy and diseased patients; most CRC studies are not) and what type of analyses are performed exactly. For this reason we have not included a quantification of these effects in the updated manuscript.

Instead, we have analyzed the effect of GC-bias correction on the estimated alpha diversity of microbial communities. We find that for most of the human gut microbiomes we analyzed, we find that the diversity is indeed underestimated when abundances are not corrected for GC-bias.

4. Second, why were only 80 samples analysed from Yachida et al? The study sequenced 616 samples. Furthermore, many more CRC microbiome datasets are available (<https://doi.org/10.1038/s41591-019-0405-7>, <https://doi.org/10.3389/fmicb.2024.1426407>). Could these be analysed as well? Is the correction of species consistent across the datasets?

We originally restricted the analysis to 80 randomly selected samples to due restrictions on available computational and storage resources. Based on the references kindly suggested by the Reviewer, we have now greatly expanded our analysis. Fig. 4 now presents data from 3506 samples from 33 studies (one of which is Yachida et al.). The large majority of these samples could be analyzed with GuaCAMOLE. As detailed in the updated manuscripts, we observe that GC-dependent sequencing efficiencies can vary widely between different gut

microbiome studies. As can be seen in the updated Fig. 4, the 33 studies we analyzed form clusters of qualitatively different types of GC-bias. Whereas the corrections are consistent within clusters, they can differ significantly between clusters.

5. *While the method is created for evaluating abundances in a metagenomics dataset, it should be possible to apply the method to a “single-genome” dataset and evaluate the efficiency of the sequencing protocol (assuming the known abundance to be 1). Would the results reflect empirically evaluated protocol efficiencies as e.g. calculated here, Figure 2 <https://academic.oup.com/gigascience/article/9/2/giaa008/5735313>?*

We downloaded the *Fusobacterium* “single-genome” datasets from the publication suggested by the reviewer (Browne *et al.*, *GigaScience*, 2020) and analyzed them with GuaCAMOLE (accessions SRR8257183, SRR8257184 and SRR8257185). Since Browne *et al.* used the *Fusobacterium* species “C1”, the genome of which is not contained in the RefSeq database, the reads got classified by Kraken2 mostly to *F. hominis*. Still, the efficiencies estimated by GuaCAMOLE reflect those reported by Browne *et al.* (Fig. A7). Notably, the bias for the NextSeq and MiSeq data is a lot stronger than what we observed in any other dataset we analyzed.

6. *False positive identification: What was the abundance of the species that were false positives? Were they low or high? One would expect that this was low abundance species, but interesting (and more important) if also higher level FP taxa were removed.*

As can be seen in Fig. A8 the removed taxa had predominantly very low abundances (with the exception of a single taxon less than 1% and on average less than 0.1%).

7. *On the mock community data the authors restrict themselves to 3 rounds of outlier removal. Was this guided by knowing the ground truth? Is the hyperparameter that gave the best value? If so it should be noted (ie. it can be overfitting when train/test is the same).*

The default value of 5 rounds of outlier removal (the previously reported number of 3 rounds was typographic error, unfortunately) is indeed based on the mock community data of Tourlousse *et al.* We have updated the Outlier Removal section to make this more clear.

8. *What is the run time and resourced required? This should also be compared against the benchmarked methods.*

We have added Table A1 which compares the runtimes of different algorithms for a sample (accession SRR12996245) containing 18.6 Million reads. The runtime is roughly on par with most other algorithms, if slightly slower. We have added a paragraph to the discussion which explains the source of this performance difference, and explains how GuaCAMOLE could be optimized by modifying Kraken2 to compute each read’s GC content.

9. In the section 6.2, the Eq (2) uses the term $P(r \in G_j|K_i)$. Even though the reference to Bracken is in place, it is unclear from the text how such probability is computed. Since it's an important part of the method, briefly describing in the text is necessary. Also, the important notations G_j and K_i are not explained. Does $P(r \in G_j|K_i)$ only apply to leaf descendants, all descendants or all descendants that have associated genomes? Eq (2) makes a summation over the whole taxonomy tree, while it seems that the non-zero probabilities can only exist for the ancestors of the node j . If that is the case, rewriting it in the terms of ancestors would improve the readability.

We have updated the text to properly define the notation used, and added a brief description explaining how $P(r \in G_j|K_i)$ is computed. Namely, by applying the Bayes' theorem to $P(K_i|r \in G_j)$, which are easily found by determining the taxa that all reads of a given length are mapped to. We have also added a sentence which makes it explicit that the sum indeed runs only over the ancestors of a node, and that $P(r \in G_j|K_i)$ is zero if K_j is not a descendent of K_i .

10. The calculations in Eq (3) are done recursively. Is it possible for the non-leaf node j to have $\tilde{O}(j, g)$ that is not equal to zero? If yes, can it lead to double counting of the number of the assigned reads?

$\tilde{O}(j, g)$ will in general be non-zero for any taxon with an associated genome, which may include internal nodes (ie. non-leaf nodes). However, the fact that $\sum_j P(r \in G_j|K_i) = 1$ ensures that the total number of reads is conserved in the redistribution process. In particular, if a read is redistributed from an internal node K_i with genome G_i to some descendant K_j with genome G_j , then that reduces the read count of K_i .

No, each read is assigned to a single taxon. In Eq (5), the term $h(j, g)$ that already aggregates over all the nodes multiplied by $P(r \in G_j|K_i)$ is used to again aggregate and multiply by the same term over all the nodes. Does it lead to double accounting of $P(r \in G_j|K_i)$, thus leading to overestimation of its effect?

In Eq. (4), the definition of $h(j, g)$, we propagate the GC distributions upwards to account for reads which are assigned to higher taxonomic nodes. In Eq. (5) we then propagate the distributions back downwards to account for read redistribution to lower taxonomic nodes by GuaCAMOLE (same as Bracken). Together, these two Equations thus partially mix the GC distributions of genomes on the same taxonomic level.

We have updated the text to make this clearer, and have also fixed a very unfortunate typo in Eq. (5) where it said $P(r \in G_i|K_j)$ instead of $P(r \in G_j|K_i)$.

12. The authors provide the test data in the repository to perform an analysis on real data. While it is an important and valuable test, it would help both the code quality and the method clarity if the repository would also have a test case with mock data. We suggest creating a mock taxonomy with 5-10 nodes, and abstract the variables that come from third-party tools, such as $P(r \in G_j|K_i)$ to

have mock values. In this case, a more clear interface between the novel method and the third-party software will be enforced and the testability and the clarity of the code will improve. Right now in order to test if the software works, one has to first build Kraken and Bracken databases. It should be possible to run a unit test without them.

We have added a folder `demo_data/` to our Github repository which contains all data necessary to run GuaCAMOLE. It contains a small pre-built Kraken2 database containing the species from the mock community of Tourlousse et al, a 1% subsample of one of their sequencing runs, and the Kraken output for the subsample. The folder contains a script which extracts all necessary data and runs GuaCAMOLE. With this, no tools other than `bzip2` are now necessary for testing GuaCAMOLE.

Minor comments

“Read counts in each taxon-GC-bin are then normalized...” - the concept of taxon-GC-bin was not introduced before, making it hard to follow the logic of the method.

We have updated the text to make the meaning of “taxon-GC” bin clearer.

“(iii) the genomic GC content distributions...” - j was not previously defined.

We have updated the text to provide a brief inline definition and have also added a reference to a later section which defines the concept of genomic GC content distributions in more detail.

Are the simulated reads from the section 6.3 also assigned to a specific taxon by running Kraken2 or the ground truth values are used?

The simulated reads used to determine $q(i, g)$ are not run through Kraken2. Instead, the miss-assignment probabilities (i.e. $P(r \in G_i | K_j)$ etc.) are used to account for that fact that reads will occasionally be assigned to higher taxonomic levels.

“Finally, we proceed similarly to Eq. (3) and average the GC distributions of all descendants, weighted by their fragment counts...” - in the Eq (6), there is an inconsistent use of γ in both numerator and denominator. In numerator, it seems like there is a typo under the nested summation, and γ is meant to be g . In denominator, γ is appearing in the two nested summation terms. Probably, the use of γ should be preferred for the nested summation term.

There were indeed multiple typographic errors in Eq. (6) which we have now corrected. γ was meant to only appear in the denominator, which represents a normalization factor that ensures that the GC distributions sum to 1.

The lines in the provided PDF are not numbered which makes it slightly more inconvenient to comment on a specific place in text.

This seems to be a property of the latex template supplied by Nature-Springer; we will see whether we can enable line numbers for our resubmission.

Reviewer 1

The paper was improved but there are still open questions:

1) Why in figure 2 we do not see all the tools that appear in figure 3? You should add comparison to all of them to figure 3.

In Fig. 2A, the goal is mainly to demonstrate that GuaCAMOLE can correctly recover different types of GC-bias, and therefore we only compare GuaCAMOLE and Bracken (which GuaCAMOLE is based on). In Fig. 2B, we had to restrict the set of algorithm to those based on RefSeq, as we explained in our previous response to comment (1) of Reviewer 1. The issue is that we lack a reliable identifier of species across different databases. While the scientific name should, in theory, suffice for this purpose, in practice it does not due to a variety of issues like slight variations in punctuation, renamed species etc. In Fig. 3, the small number of species allowed us to resolve such issues manually. However, for the simulations in Fig. 2B this was not feasible, and we were thus unable to guarantee a fair comparison with algorithms not based on RefSeq. Consequently, we chose to omit these algorithms from the comparison in Fig. 2B. Since the main point of Fig. 2B is to demonstrate that GuaCAMOLE works for communities of a wide range of complexities, we believe this choice to be sound.

2) It is hard to re-review the paper without marking the new changes. You should provide a version with track changes relatively to the first version.

We apologize for this issue and any inconvenience it may have caused.

3) Need to better discuss in the discussion section the cases that the suggested approach fails (e.g. for very small communities whose taxa all have extreme GC content). Is there a way to fix this ?

Since GuaCAMOLE relies on the overlap between the GC distributions of individual taxa to estimate sequencing efficiencies, not much can be done in the case of insufficient overlap. However, GuaCAMOLE does attempt to detect this issue, and will warn the user and refuse to report efficiency estimates if this is detected. We have amended the discussion to explain this in more detail.

4) Need to provide in the supplementary the raw data used to generate the figures (e.g. figure 2 – what are the genomes used)

We agree with the reviewer, and have made all raw data and scripts necessary to reproduce figures available at <https://github.com/Cibiv/GenomicGC-BiasCorrectionImprovesAbundanceEstimation>. In particular, this repository contains in `simulation/parameters` one file for each simulate community which lists the assemblies present (in the form of a RefSeq accession) and their abundances.

5) In Fig. 2B the plots of the 3 algorithms are not on the same x-axis value: is this due to visualization reasons (in this case, it should be explained in the caption) or is it a real phase in the number of taxa (in this case, it should be fixed).

We thank the Reviwer for pointing out this omission in the figure legend, and have added the sentence “To avoid overlaps, dots are horizontally shifted (GuaCAMOLE left, MetaPhlAn4 right) and jittered.” to clarify.

6) I expect to see more points in each group in 2B.

In Fig. 2B the main goal is to test the performance of GuaCAMOLE for communities of different complexity and distributions of genomic GC content. We believe that the figure in its current version achieves the desired goal: It shows that GuaCAMOLE can be expected to reduce the average estimation error in all cases except that of small communities comprises of species with very extreme GC content. Additional simulations are, in our opinion, unlikely to alter this conclusion.